# Survey on Reporting of Child Abuse by Pediatricians: Intrapersonal Inconsistencies Influence Reporting Behavior More than Legislation

**DOI:** 10.3390/ijerph192315568

**Published:** 2022-11-23

**Authors:** Oliver Berthold, Vera Clemens, Benjamin H. Levi, Marion Jarczok, Jörg M. Fegert, Andreas Jud

**Affiliations:** 1Department of Child and Adolescent Psychiatry/Psychotherapy, University of Ulm, Steinhövelstr. 5, 89075 Ulm, Germany; 2Departments of Humanities and Pediatrics, Penn State College of Medicine, 700 HMC Crescent Road, Hershey, PA 17033, USA; 3School of Social Work, Lucerne University of Applied Sciences and Arts, Werftestrasse 1, 6002 Lucerne, Switzerland

**Keywords:** reasonable suspicion, mandatory reporting, child abuse, decision making, maltreatment

## Abstract

**Background**: Internationally, various laws govern reporting of child abuse to child protection services by medical professionals. Whether mandatory reporting laws are in place or not, medical professionals need internal thresholds for suspicion of abuse to even consider a report (“reasonable suspicion” in US law, “gewichtige Anhaltspunkte” in German law). **Objective:** To compare internal thresholds for suspicion of abuse among US and German pediatricians, i.e., from two countries with and without mandatory reporting laws. **Participants and Setting:** In Germany, 1581 pediatricians participated in a nationwide survey among child health professionals. In the US, a survey was mailed to all Pennsylvania pediatricians, and 1249 participated. **Methods:** Both samples were asked how high in their rank order of differential diagnoses child abuse would have to be when confronted with a child’s injuries to qualify for reasonable suspicion/gewichtige Anhaltspunkte (differential diagnosis scale, DDS). In a second step, both had to mark a 10-point likelihood scale (0–100%) corresponding to reasonable suspicion/gewichtige Anhaltspunkte (estimated probability scale, EPS). **Results:** While for almost two-thirds of German pediatricians (62.4%), child abuse had to be among the top three differential diagnoses for gewichtige Anhaltspunkte, over half of the US respondents (48.1%) had a lower threshold for reasonable suspicion. On the estimated probability scale, over 65% in both samples indicated that the probability of abuse had to exceed 50% for reasonable suspicion/gewichtige Anhaltspunkte. There was great variability between the two countries. **Conclusions:** There are similar uncertainties in assessing cases of suspected child abuse in different legal systems. There is a need for debates on thresholds among medical professionals in both countries.

## 1. Introduction

Given the large prevalence of child maltreatment [1] and its largely devastating consequences for both the individual and society [2,3], most countries passed laws governing reporting of suspected child maltreatment to child and youth welfare authorities by professionals in contact with children, including medical professionals. However, the legal frameworks are quite variable, as are arguments regarding the pros and cons of mandatory reporting [4,5,6,7]. In Germany, the vast majority of pediatricians and child psychiatrists report feeling a moral obligation to involve child protection services [8]. However, while mandatory reporting laws exist in the US and Canada, Germany (along with several other European countries) does not require medical professionals to report suspected child maltreatment to child and youth welfare. To the extent that there is common ground between Germany and other countries with regard to reporting suspected child maltreatment, it involves the threshold that is set for taking action (i.e., making a report). In many English-speaking countries, “reasonable suspicion” serves as the standard threshold for mandated reporting [9]. In German law, a report is warranted if it meets the threshold of “gewichtige Anhaltspunkte”, whose literal translation is “weighty evidence”. Much like “reasonable suspicion”, “gewichtige Anhaltspunkte” is intended to convey that there must be some element of gravity, seriousness, or weight that attaches to one’s concern and that reports should not be made on the basis of just hints or mere suspicions.

Even though US and German law use fundamentally different legal frameworks, practically speaking, both use thresholds with a similar intent for when a child abuse report should be made. To promote appropriate intervention in cases of child maltreatment, consistent understanding of this threshold by medical professionals is crucial [10]. Whether a particular case carries a low, medium, or high likelihood of actually being maltreated is a determination that ultimately rests with highly trained professionals, and there is a substantial body of research to assist in this decision-making process [11,12,13,14]. However, inter-rater variability is high even among experts in their respective fields [15]. It is for this reason that well-trained interdisciplinary child protection teams using an evidence-based approach are more likely to make decisions that improve the well-being of children when maltreatment is suspected [16,17]. By contrast, prior research with individual medical professionals in the US showed considerable variability and inconsistency in how they make decisions about possible child maltreatment [9,10,18,19]. 

This study examines how German pediatricians, child psychiatrists, and child psychotherapists understood the legal threshold for reporting possible child maltreatment and compared it to previously published data from the US [9,10]. Using the same research instruments developed by Levi et al. [9,10], we hypothesized that medical professionals in Germany would demonstrate similar variability and inconsistency despite the different legal frameworks.

## 2. Materials and Methods

### 2.1. Samples and Procedures

**Germany.** The German sample is a part of a larger survey of pediatricians, child and adolescent psychiatrists, pediatric surgeons, and psychotherapists. Members of these professional associations were sent e-mails, and the survey was also advertised in their respective newsletters and websites. As an incentive to participate, tickets for the Congress of the German Society of Pediatrics and Adolescent Medicine were raffled. For this comparison, we analyzed only the subgroup of pediatricians. Age and gender distributions of the various groups surveyed are shown in Table 1. Out of 18,102 contacted physicians, 1581 pediatricians participated (response rate = 9.1%). Participating pediatricians were slightly younger compared to the mean age of members of the associations (43.2 years). There was a slightly significant gender difference between participants of the survey and the overall gender distribution of the medical associations, but this was not significant (female participants: 67.6%, female members of associations: 64.6%, *p* = 0.053).

**United States.** The study that was previously conducted in the US state of Pennsylvania included only licensed pediatricians and had a 60% response rate, totalling 1249 participants [9,10]. In that study, hard copies of the survey were mailed to all licensed pediatricians (along with a USD 5 bill to incentivize completion of the survey), and the demographics of respondents are also shown in Table 1.

### 2.2. Instruments and Operationalization

The two central instruments used in the present study of German medical professionals were translated into German but were otherwise identical to those used in Levi et al.’s prior research [9,10]. The **Differential Diagnosis Scale (DDS)** asks respondents to mark on a rank-ordered list (where 1 is the most likely diagnosis, and 10 is the least likely diagnosis) how likely (suspected) child maltreatment would have to be as a potential diagnosis to qualify as reasonable suspicion/gewichtige Anhaltspunkte. The DDS was analyzed as one of the dependent variables. The **Estimated Probability Scale (EPS)** was used as our second dependent variable. On the EPS, respondents were asked to mark the point on a 10 cm scale (ranging from 0% likelihood to 100%, i.e., certainty) corresponding to the likelihood they thought was necessary to constitute reasonable suspicion/gewichtige Anhaltspunkte. Because the German survey was completed online, respondents marked the threshold via a mouse click (rather than the paper/pencil version used in the US).

Since both German and US legislation use the umbrella term of maltreatment and do not limit it to certain forms, we have also used the general term in our study. 

**Gender** was operationalized as either female or male due to only 8 respondents (0.4%) in the German survey identifying their gender as “other” (which was too small a group to render meaningful comparison data), and their data were excluded from further analyses. **Age** of the respondents was categorized into three nominal categories, less than 41 years, 41–60 years, more than 60 years. **Education on child maltreatment** was binary coded and operationalized for the German survey as having attended a continuing education (course, seminar) on child maltreatment and for the US survey as having prior “education or guidelines as to what constitutes child abuse.” Finally, the **caseload** in relation to child maltreatment was operationalized for the German survey as having any case involving suspected child maltreatment in the last 12 months and for the US survey as having reported cases of alleged maltreatment in the last 24 months. This difference in operationalizing caseload reflects the different legal frameworks, with the US mandating medical professionals to report but no such mandate in Germany. To compare the two and account for the difference in time frames, caseload for the US sample was divided by 2.

### 2.3. Data Analyses

In addition to descriptive analyses, associations of dependent variables (rating a case as reasonable suspicion) with predictors (gender, age, education on maltreatment, caseload) have been analyzed using multivariate ordinal logistic regressions (DDS) or multivariate linear regressions (EPS).

## 3. Results

### 3.1. Comparison of German and US Pediatricians

In the German sample (Figure 1), almost two-thirds of respondents (63.0%) indicated that child maltreatment had to be one of the top three diagnoses on the DDS for there to be reasonable suspicion/gewichtige Anhaltspunkte, compared to only about half of the US respondents (48.1%). A larger number of US pediatricians (22.7%) compared to their German colleagues (12.4%) indicated that a ranking below five on the DDS could qualify as reasonable suspicion/gewichtige Anhaltspunkte of child maltreatment. That said, the clear majority of medical professionals in both countries indicated that maltreatment should be within the top five possible diagnoses for there to be reasonable suspicion/gewichtige Anhaltspunkte (87.6% of the German and 77.3% of the US sample). 

On the EPS, the response from German medical professionals demonstrated a crescendo/decrescendo pattern (Figure 2), with the largest subgroup (21.0%) indicating that at least an 80% probability (of child maltreatment) was needed to qualify as gewichtige Anhaltspunkte. This contrasts with the almost normal distribution seen in the US sample, where the largest subgroup (25.30%) indicated that a probability of at least 50% was needed to qualify as reasonable suspicion. That being said, 78.2% of respondents in the German sample and 68.5% in the US sample indicated that an estimated probability of 50% or greater was needed in order for reasonable suspicion/gewichtige Anhaltspunkte to exist. 

### 3.2. Predictors of Response to the Differential Diagnosis Scale and the Estimated Probability Scale

For the DDS, there were few strong predictors for the threshold that individuals identified for reasonable suspicion/gewichtige Anhaltspunkte to be present (Table 2). However, in the German sample, female gender and higher age were significant predictors for assuming a lower probability for abuse (equivalent to having a higher threshold for a report), while in the US, there were no significant predictors for assuming a high probability. For the EPS, there were no significant predictors of response in the German sample (Table 3). In the US sample, the only predictors were having higher levels of expertise and/or prior education in child abuse, both of which were correlated with lower thresholds for the probability needed to qualify as reasonable suspicion.

Both samples showed high heterogeneity in determining the likelihood of child maltreatment that would warrant a report to child protection. In both samples, not only was there high variability between different professionals with similar training, but also for the same individual for the two different scales. Logically speaking, an estimated probability of abuse over 50% must necessarily correspond to a first position on the differential diagnostic scale. This is because each of the individual likelihoods for the various explanations of a child’s condition cannot add up to more than 100% probability.

The linear regression results show a weak relationship (in the anticipated direction) between these two measures (R^2^ = 31.3%) in the combined sample (Figure 3). Moreover, the variance is significantly more pronounced for the German pediatricians than for their American colleagues (LR test for interaction *p* < 0.0001). This is especially evident with decreasing values on the EPS, as these German pediatricians assigned significantly higher position on the DDS list. Separate models indicate a slope of b = 0.03 and R^2^ = 12.3% for Germany and a slope of b = 0.06 and R^2^ = 35.9% for the US (see regression lines in Figure 3). The variability between pediatricians in one country was greater than the variability between the two different samples from Germany and the US (Figure 3).

## 4. Discussion

For policymakers, it is crucial to understand how the goal to facilitate reporting of suspected child abuse by medical professionals can be achieved and which effects mandatory reporting laws have on the professionals involved.

To the best of our knowledge, this is the first comparison of how medical professionals working with children in Germany and the US understand and operationalize the concept of reasonable suspicion/gewichtige Anhaltspunkte.

Overall, German medical professionals seem to apply significantly higher thresholds for reasonable suspicion/gewichtige Anhaltspunkte compared to the US sample. One reason may lie in the different systems. Although the German system encourages reporting, it does not mandate it—contrary to the US system—which may send the message to German medical professionals that the likelihood of child maltreatment should be fairly high to trigger a report.

However, the results also indicate that in both systems, intra- and inter-rater reliability in the assessment of reasonable suspicion of child maltreatment is low. Prior work by Levi et al. argues that to be logically consistent, any score ≥50% on the scale of estimated probability would need to correspond with a ranking of 1 on the differential diagnosis score, an estimated probability of ≥34% would correspond with a ranking no lower than differential diagnosis score of 2 and an estimated probability of ≥25% no lower than a ranking of 3 on the differential diagnosis score [9]. However, in our sample (as with prior research), the vast majority of respondents indicated on the EPS that reasonable suspicion/gewichtige Anhaltspunkte required an estimated probability of 50% or above, but the vast majority also indicated that a rank of 2–5 on the DDS was sufficient to count as reasonable suspicion/gewichtige Anhaltspunkte. This inconsistency held true for individuals as well as across groups. This means that a single professional can come to completely different results in the assessment of similar cases and that psychotherapists can come to a different conclusion than pediatricians when assessing similar cases. Interestingly, this phenomenon was consistent between the two countries, especially in the high likelihood of abuse. When lower ranks on the EPS were allocated, US pediatricians were more likely to rank abuse higher on the DDS than their German colleagues. The inconsistency was more pronounced, and the lower abuse was ranked on the EPS. Together, these findings raise concerns about how individuals interpret the threshold for reporting suspected child maltreatment. Earlier research on physicians and nurses that showed high levels of uncertainty when only one medical professional was involved [9,10,18,19] may help explain why child maltreatment often goes undetected in the health care system [20]. Given that about one-third of primary care pediatricians in the German state of Berlin reported that they do not see any cases of suspected child physical maltreatment [8], one must wonder how many children are being left in harm’s way because of how they interpret the threshold for reporting possible maltreatment. This is all the more troubling, as population surveys on the epidemiology of child maltreatment in Germany point to high prevalence rates. In a 2016 nationwide sample, 31% of respondents reported having experienced at least one form of maltreatment as a child [21].

It could be argued that one main advantage of mandatory reporting is clarity: if professionals know that they have to report, they might do so more readily. However, the variability of the present data shows that medical professionals in both legal systems struggle to interpret the threshold for reporting in a way that is consistent. It is an open question whether such an interpretation would be more consistent/reliable if it was carried out by multi-disciplinary groups. The advantage of enlisting other professionals with different perspectives and training is that it encourages deliberation and, thereby, critical thinking. In both ambulatory and hospital care, involving multi-disciplinary child protection teams has been shown to improve the handling of child protection cases [16,17]. However, whether this is scalable is an empirical question.

It seems that the legal framework is not the key factor to comprehensive thresholds of decision-making in suspected child abuse. Whether there is mandatory reporting or not, continuous education and counsel on guidelines and multi-professional teams might help improve clinical decision making in suspected child abuse.

One major limitation of the study is that in both samples, the responses may not be representative of the entire medical profession, particularly given the limited participation in the German online survey. That said, it is reasonable to think that any selection bias in participants actually skews toward higher participation by professionals motivated to promote child protection. As such, the significant variation seen in the present findings may well increase in a broader sample of participants. Another limitation of this comparison of data is the different time frames, with the US data collected in 2005 and the German data collected in 2019/2020.

Reasonable suspicion and gewichtige Anhaltspunkte are two closely similar, but not equivalent undefined legal terms, i.e., they both have to be interpreted by physicians. It is not possible to disentangle variance between the two concepts due to differences in language from the likely larger variance due to the legally undefined nature of the concepts. Finally, it must be acknowledged that there are numerous factors that influence reporting, separate from how a person interprets reasonable suspicion/gewichtige Anhaltspunkte [16].

Furthermore, we are aware that there are overarching, cultural differences between the countries involved. for example, the attitude of a society to corporal punishment in general. A closer examination of these differences was not part of this study; as legal and social-cultural differences between the two settings co-exist, we have not been able to disentangle these two contributors. Finally, we cannot rule out a time effect as the survey among US pediatricians was performed 15 years before the survey among German health care professionals. Yet, reporting legislation has not changed in the US setting.

## 5. Conclusions

The present data provide important insight into thresholds that pediatricians use for deciding when to report cases that may involve child maltreatment. Results indicate that in both Germany and the US, medical professionals struggle to interpret the threshold for reporting in a way that is consistent. This inconsistency can result in the failure to protect maltreated children, as well as the misallocation of limited child welfare resources. There is a need for a transparent debate on thresholds of reasonable suspicion among medical professionals in Germany and the US in order to enable consistent, effective child protection.

## Figures and Tables

**Figure 1 ijerph-19-15568-f001:**
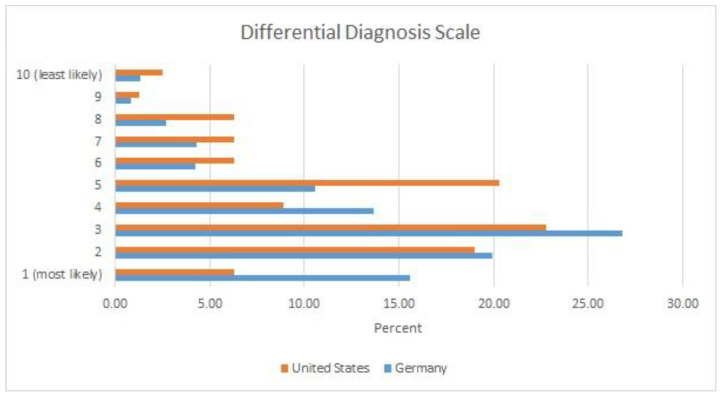
Differential Diagnosis Scale. Answers to the question: “Where on your own list of differential diagnoses should child abuse be located for you to consider reasonable suspicion/gewichtige Anhaltpunkte for abuse?”.

**Figure 2 ijerph-19-15568-f002:**
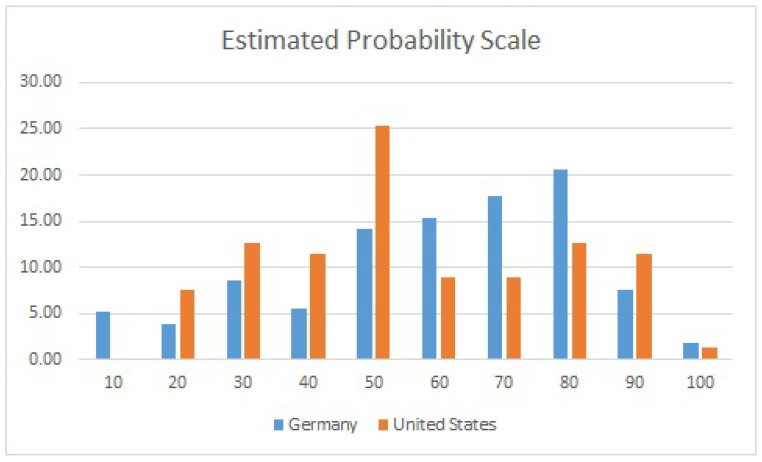
Estimated probability scale. Answers to the question: “What likelihood of child abuse (0–100%) would be necessary to have “reasonable suspicion”/“gewichtige Anhaltspunkte” of abuse?”.

**Figure 3 ijerph-19-15568-f003:**
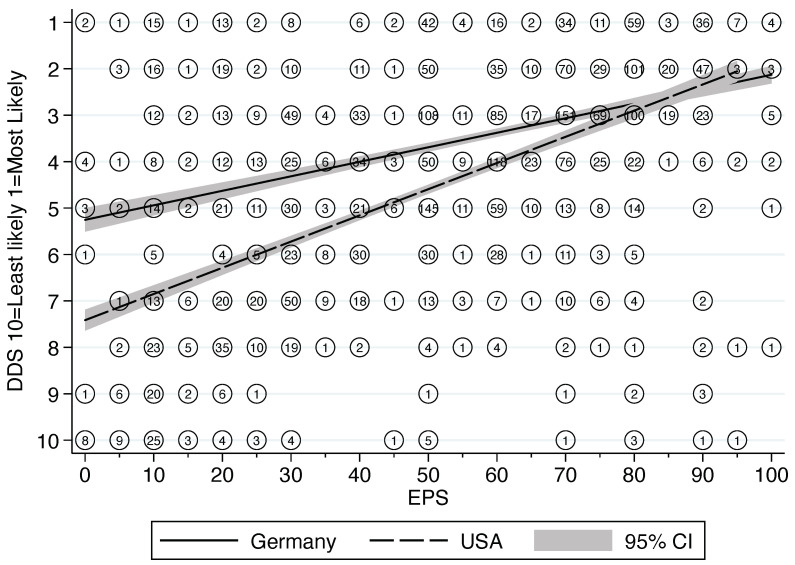
Linear regression of the relationship between differential diagnosis scale and estimated probability scale. Comparison of pediatricians from the US and Germany.

**Table 1 ijerph-19-15568-t001:** Samples.

	Germany	United States
Total sample	1581	1249
Percentage female ^1^	67.6%	55%
Age		
<41 years	41.9%	46.6%
41–60 years	46.7%	41.8%
>60 years	11.4%	10.6%

^1^ Note: In Germany, 8 participants (0.51%) identified their gender as “other” and were excluded from further analyses.

**Table 2 ijerph-19-15568-t002:** Predictors for reasonable suspicion using a differential diagnosis scale.

Predictor	Germany				United States			
	Odds Ratio ^1^	z-Score	95% Confidence Interval		Odds Ratio ^1^	z-Score	95% Confidence Interval	
Female gender	0.75	−2.77 **	0.617	0.920	1.22	1.91	0.99	1.48
Age 40–60 years	0.87	−1.40	0.718	1.057	1.14	1.24	0.93	1.41
Age >60 years	0.51	−4.06 ***	0.373	0.709	1.05	0.25	0.74	1.48
Education on CM	0.95	−0.51	0.771	1.165	0.87	−1.36	0.71	1.07
1–5 cases	1.10	0.79	0.868	1.398	0.91	−0.81	0.73	1.14
6+ cases	1.07	0.47	0.798	1.449	0.82	−1.13	0.59	1.16

^1^ An OR > 1 corresponds to a higher probability of assessing the situation as reasonable suspicion; comparison categories from top to bottom: Male gender, age group < 41 years, 0 child protection cases, no child protection training; ** *p* < 0.01; *** *p* < 0.001.

**Table 3 ijerph-19-15568-t003:** Predictors for reasonable suspicion using an estimated probability scale. Differential Diagnosis and Estimated Probability.

Predictor	Germany				United States			
	Coefficient	z-Score	95% Confidence Interval		Coefficient	z-Score	95% Confidence Interval	
Female gender	2.07	1.75	−0.248	4.386	−0.55	−0.42	−3.14	2.04
Age 40–60 years	0.86	0.75	−1.400	3.118	1.41	1.02	−1.31	4.14
Age >60 years	−1.49	−0.79	−5.188	2.202	1.29	0.55	−3.30	5.89
Education on CM	−2.13	−1.80	−4.459	0.197	−2.05	−1.51	−4.72	0.61
1–5 cases	−0.59	−0.41	−3.414	2.228	−2.18	−1.50	−5.02	0.67
6+ cases	−3.17	−1.86	−6.509	0.175	−5.46	−2.42 **	−9.89	−1.03

Note: Comparison categories from top to bottom: Male gender, age group < 41 years, 0 child protection cases, no child protection training; ** *p* < 0.01.

## Data Availability

The data presented in this study are available on request from the corresponding author. The data are not publicly available due to privacy requests of the participating professional associations.

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
