# Peer review of "Survey on Reporting of Child Abuse by Pediatricians: Intrapersonal Inconsistencies Influence Reporting Behavior More than Legislation"

_ijerph, 2022, doi:10.3390/ijerph192315568_

Round 1

Reviewer 1 Report

Lines 208-209. What do you think is due? could you explain more extensively?

Have you considered the cultural differences between both samples?

The term maltreatment is wide and includes different types. Could you explain on what types the differential diagnosis is made?

Have you performed statistical analyzes on gender differences in decision-making on differential diagnosis? Do you think it could be relevant?

Also, I am not sure that the authors have made the limitation of the different time-frames between the two samples clear enough. why was it used in this study?

Author Response

Thank you for your valuable comments to our manuscript. Please find our point-to-point reply below:

  • We have adapted the wording to make our point clearer (line 211).
  • With cultural differences, either the professional culture could be meant or a social one. The first is framed in particular by the legal settings we work in, thus i.e. whether there is mandatory reporting in cases of suspected child abuse. We have dealt with this in detail. On social-cultural aspects, we do not have any data in the respective samples. Moreover, legal and social-cultural differences co-exist, the two sets of contributors cannot be disentangled. We have therefore added this aspect to the limitations section (line 257).
  • In our study, we leaned on the terms used in the German and US legislation – there, all forms of maltreatment are included. Therefore, we did not specify this in our study either. We added an explanation to line 111 to make this point clearer.
  • We believe that gender differences are highly relevant in decision making processes in cases of suspected maltreatment. However, to our surprise, the difference in our sample was not statistically significant (see line 85).
  • We added this limitation to line 262.

Reviewer 2 Report

This article is interesting and well-constructed. Some details are necessary in the results and discussion section to better understand this study. My comments are in the below:

1.Fig. 3?

2.In the discussion,Line 193, One 193 reason may lay in the different systems. How different systems compare U.S. and German surveys of child abuse Reports?

Author Response

Thank you for your valuable comments to our manuscript. Please find our point-to-point reply below:

  • The figures seem to have been missing – we’ll resubmit them with the revised manuscript.
  • Regarding line 193 – we agree that different systems, particularly legal frameworks, influence reporting behavior. We clarified intra- and interrater variability in line 211 and added the issue to the limitations (line 259).

Reviewer 3 Report

I would like to thank the opportunity of review the article titled “Survey on reporting of child abuse by pediatricians: Intrapersonal inconsistencies influence reporting behavior more than legislation”. It was a very interesting work on how US and German pediatricians understood the legal threshold for reporting child maltreatment. The method is well documented, and the analysis is strong. I found both results and discussion very relevant and with enough references. Because of that, I would like to congratulate the authors for doing such a good job.

The only question that comes to mind is about the text on lines 133, 143, 177 and 184, when authors refer to Figures 1 to 3 respectively. However, there are no figures in the text.

Author Response

Thank you for your valuable feedback! We apologize that the figures seem to have been missing. We'll upload them with the revised manuscript.

Round 2

Reviewer 1 Report

I have made the new review of the manuscript, and I accept the new modifications and clarifications made by the authors.